

# A SE-DenseNet-LSTM model for locomotion mode recognition in lower limb exoskeleton

Jing Tang[1,2], Lun Zhao[1], Minghu Wu[1,2], Zequan Jiang[1], Jiaxun Cao[1] and Xiang Bao[1]

[1] Hubei Key Laboratory for High-efficiency Utilization of Solar Energy and Operation Control of Energy Storage System, Hubei University of Technology, Wuhan, China
[2] Hubei Engineering Research Centre for Safety Monitoring of New Energy and Power Grid Equipment, Hubei University of Technology, Wuhan, China

## ABSTRACT

Locomotion mode recognition in humans is fundamental for flexible control in wearable-powered exoskeleton robots. This article proposes a hybrid model that combines a dense convolutional network (DenseNet) and long short-term memory (LSTM) with a channel attention mechanism (SENet) for locomotion mode recognition. DenseNet can automatically extract deep-level features from data, while LSTM effectively captures long-dependent information in time series. To evaluate the validity of the hybrid model, inertial measurement units (IMUs) and pressure sensors were used to obtain motion data from 15 subjects. Five locomotion modes were tested for the hybrid model, such as level ground walking, stair ascending, stair descending, ramp ascending, and ramp descending. Furthermore, the data features of the ramp were inconspicuous, leading to large recognition errors. To address this challenge, the SENet module was incorporated, which improved recognition rates to some extent. The proposed model automatically extracted the features and achieved an average recognition rate of 97.93%. Compared with known algorithms, the proposed model has substantial recognition results and robustness. This work holds promising potential for applications such as limb support and weight bearing.

## INTRODUCTION

Exoskeleton robots have been widely and effectively utilized in diverse fields of medical rehabilitation, military training, and civilian scenarios (*Zhang et al., 2022*). Currently, lower limb exoskeletons serve as assistive devices for individuals with disabilities, functioning similarly to medical robots. In addition, they also help able-bodied individuals enhance physical strength as load-bearing or power-aiding exoskeletons (*Shi et al., 2019*). The recognition of human motion intent is mainly divided into recognizing locomotion modes and detecting gait phases (*Zheng et al., 2022b*). Locomotion mode recognition is considered the central focus of research for motion intent recognition, which is in line with this research. Accurately recognizing human locomotion modes is fundamental for

Corresponding author
Minghu Wu, 102110397@hbut.edu.cn

exoskeleton robots to interact seamlessly with the human body. With this capability, the exoskeleton can predict and respond to human motion, and better adapt to the motion intent of individuals. Therefore, it is crucial to recognize human locomotion modes accurately in exoskeleton robots.

So far, various methods have been utilized to capture human motion signals to recognize human locomotion modes. For instance, bioelectric signals (*Kumari, Mathew & Syal, 2017*), visual-based (*Liu et al., 2019*; *Beddiar et al., 2020*), and mechanical sensors (*Zheng et al., 2022a*; *Xia, Huang & Wang, 2020*) have been applied in this field.

The method based on bioelectric signals is mainly collected by surface electromyography (EMG) (*Wilcox et al., 2016*; *Meng et al., 2021*; *Vijayvargiya et al., 2022*) and electroencephalography (EEG) (*Chaisaen et al., 2020*; *Zhou & Gao, 2021*). This approach boasts low latency and enables the direct acquisition of human motion intent. However, signals collected in this way can be influenced by external factors, such as variations in the wearer's body size, changes in body temperature, or the presence of perspiration. Meanwhile, bioelectric sensors can be complicated to wear, which has significant limits in practical applications (*Wang et al., 2022*).

Vision-based methods generally capture images of individuals in motion through a stationary camera, which are then analyzed to identify respective locomotion modes. This approach can effectively enhance the accuracy of recognizing various locomotion modes. Nevertheless, it could potentially violate personal privacy and become susceptible to external factors, such as light intensity and background changes. Consequently, the utilization of vision-based recognition methods is limited by these potential problems (*Zhang et al., 2017*; *Abu-Bakar, 2019*; *Singh & Vishwakarma, 2019*).

The mechanical sensor-based approach (*Han, Wong & Murray, 2019*; *Wu et al., 2019*; *Semwal, Gupta & Lalwani, 2021*) is another viable approach for recognizing locomotion modes in lower limb exoskeleton robots. This method uses inertial measurement units (IMUs), force sensors, and angle sensors for data gathering. The data acquisition device used for mechanical sensors is convenient to wear and suitable for a wide range of environments. Moreover, as a non-invasive mechanical sensor, IMU can capture small motion changes and details, which is more widely used. *Liu et al. (2017)* devised a mechanical sensing system for human locomotion modes. IMUs and plantar pressure sensors were incorporated into the system to compensate for the absence of EMG signals, resulting in improved recognition rates. Therefore, the mechanical sensor-based is an effective data acquisition system to recognize locomotion modes in exoskeleton robots.

The methods of machine learning have been traditionally utilized for recognizing locomotion modes in lower limb exoskeletons, for example, K-nearest neighbor (KNN) (*Cheng, Bolívar-Nieto & Gregg, 2021*), and support vector machine (SVM) (*Fei et al., 2020*). These methods can improve recognition rates by manually extracting features when dealing with limited amounts of data. However, performing computations through traditional methods machine of learning can be slow when handling datasets containing large amounts of information (*Iqbal et al., 2021*). Especially in complex locomotion mode recognition tasks, meeting the requirements of real-time processing and high computational complexity can present a challenge.

As artificial intelligence continues to advance, the utilization of deep learning is becoming increasingly common in human locomotion mode recognition. To address the time-consuming task of manual feature extraction by model-based methods, *Zheng et al. (2022b)* combined convolutional neural network (CNN) and SVM with a finite state machine (FSM) to extract automatically human information features collected with IMUs, which recognize five single and eight mixed locomotion modes with recognition rates of 97.91% and 98.93%. Similarly, *Wang et al. (2022)* presented a method that adaptively combined a genetic algorithm (GA) with CNN. It recognized twelve locomotion modes with high accuracy and low latency through multi-sensor information selection. *Xia, Huang & Wang (2020)* introduced a deep neural network incorporating both convolutional layers and long short-term memory (LSTM), which achieved high recognition rates of 95.78%, 95.85%, and 92.63% on three public datasets. *Mohsen, Elkaseer & Scholz (2021)* used a hybrid model combining CNN with LSTM (CNN-LSTM) to analyze a dataset collected from 36 individuals performing different locomotion modes. It trained using the TensorFlow framework and tuning hyperparameter methods to achieve high accuracy. Additionally, *Xu et al. (2019)* presented a deep learning model which integrates recurrent neural networks and gated recurrent unit (GRU), effectively extracting time series features from the data. *Zhu et al. (2020)* developed a model that combined Dense Convolutional Network (DenseNet), LSTM, and multilayer perceptron (MLP), resulting in lower switching scene time delays and higher recognition rates.

A SE-DenseNet-LSTM hybrid model merging multi-dimensional data is proposed. The features between the layers of the network are sufficiently explored by constructing a DenseNet model. Furthermore, LSTM can selectively forget and update information through the introduction of the gating mechanism and the dropout mechanism. Therefore, the LSTM layer is adept at capturing long-term dependence within the time series, while concurrently mitigating the problem of overfitting in the model. To tackle the challenge of poorly characterized ramp data, the SENet module is introduced. The experimental results indicated that the proposed hybrid model improved recognition rates for various locomotion modes while exhibiting better generalization performance. The following are the principal contributions of this work:

- Using IMUs to collect gait data in different terrains, which is used to recognize five locomotion modes, such as level ground walking, stair ascending, stair descending, ramp ascending, and ramp descending.
- Designing a hybrid DenseNet-LSTM model with an attention module, which possesses higher recognition rates compared with SVM, CNN, LSTM, etc.
- Using the hybrid SE-DenseNet-LSTM model to effectively tackle the challenge presented by inconspicuous features of ramp data.

The rest of this article is organized as follows. The "Materials & Methods" section introduces a comprehensive account of the data acquisition system, the specific content of DenseNet, LSTM, and the hybrid models. The "Experimental Results and Discussions" section presents the results and analysis of this research, and contrasts them with previous algorithms while providing a discussion. "Conclusions" summarizes this research.

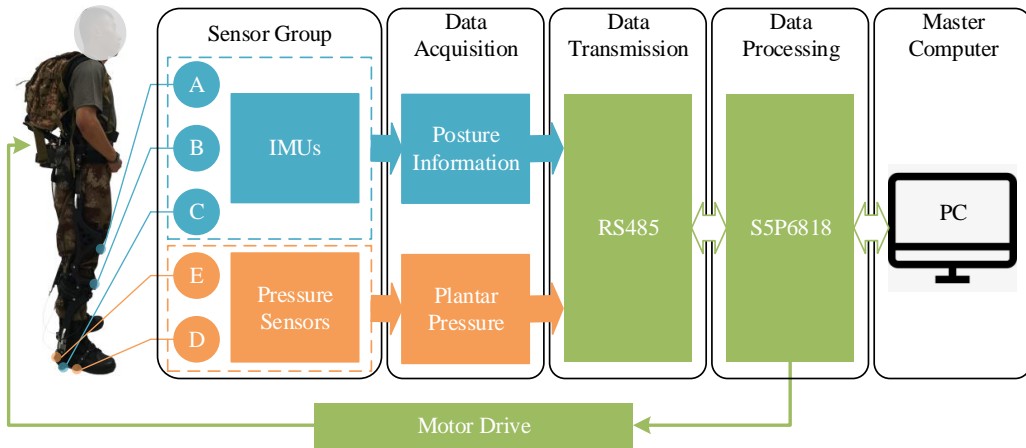

**Figure 1** **Mounting locations of IMUs and pressure sensors in lower limb exoskeletons.**

## MATERIALS & METHODS

### Data acquisition system

IMUs and pressure sensors are used to construct the data acquisition system in Fig. 1. IMU is utilized for measuring both the orientation and acceleration of the subject being monitored, while the pressure sensor is used to measure the pressure changes in the surrounding environment. Specifically, the wearable exoskeleton is mounted with individual IMUs (MPU9250) on thighs (A), shanks (B), and feet (C) to accurately determine the angles of the hip, knee, and ankle from the received data. In addition, pressure sensors (LOSON LSH-10) are also mounted in the positions of the soles (D) and heels (E) to get gait information. The sampling frequency is 100 Hz. To tackle the problem of data latency resulting from the wireless transmission, the wired transmission method (RS485 bus) is used. The main chip in the exoskeleton circuit is an octa-core processor (S5P6818) equipped with Samsung Cortex-A53, using 2GB RAM and 8GB EMMC.

Motion data were collected from 15 subjects (height: 177 ± 8 cm, weight: 66 ± 6 kg, and age: 33 ± 8 years) wearing an exoskeleton with no disease. The experiments were carried out on three different terrains: flat, stairs, and a 10° ramp. Sensors on the exoskeleton were carefully inspected and calibrated to ensure accurate data collection. This study was approved by the Experimental Ethics Committee of Exercise Science of Beijing Sport University (Ethical Application Ref: 2019007H). Written and verbal consent was obtained from each study participant.

Furthermore, the joint portion of the exoskeleton was also customized to accommodate the natural motions of individual subjects. All subjects wore the exoskeleton for pre-adaptation, walking at normal speed over each of the three terrains for approximately one minute. In addition, the exoskeleton was continuously fine-tuned to ensure maximum comfort and dexterity for the subjects. Subjects performed experiments at a constant walking speed (3 km/h), including five locomotion modes of level ground walking (FW),

stair ascending (SA), stair descending (SD), ramp ascending (RA), and ramp descending (RD) in three terrains.

## Data pre-processing

During the data collection, the angle of the ankle joint was correlated with the walking habits of different subjects, resulting in unstable fluctuations in the angle cycle of the ankle joint. Therefore, the hip and knee angles, along with IMU signals from the shank, are selected as input signals to the neural network and represented as

$$\text{Input} = [\theta_{LH}, \theta_{LK}, \theta_{LS}, \theta_{RH}, \theta_{RK}, \theta_{RS}] \tag{1}$$

Where $\theta$ represents the angle and the subscripts LH and RH, LK and RK, and LS and RS represent the right and left hip, knee, and shank. The given angles are defined in Fig. 2.

$\theta_{\text{Hip}}$, $\theta_{\text{Knee}}$, and $\theta_{\text{Ankle}}$ represent the joint angle of the hip, knee, and ankle. In which, $\theta_{\text{Hip}}$ is the angle formed by the intersection of the vertical axis with the thigh. When the thigh is bent forward, $\theta_{\text{Hip}} > 0$; when it is bent backward, $\theta_{\text{Hip}} < 0$. Similarly, $\theta_{\text{Knee}}$ is the angle formed by the intersection of the shank direction with the thigh extension. $\theta_{\text{Ankle}}$ is the angle formed by the intersection of the plumb line in the plane of the sole with the shank. Specifically, the representation is as

$$\begin{cases} \theta_{\text{Hip}} = \theta_{\text{Thigh}} \\ \theta_{\text{Knee}} = \theta_{\text{Shank}} - \theta_{\text{Thigh}} \\ \theta_{\text{Ankle}} = \theta_{\text{Foot}} - \theta_{\text{Shank}} \end{cases} . \tag{2}$$

Curves of motion data were plotted based on motion information from subjects while walking. All joint angles and plantar pressures were cyclically variable in the lower limbs. The motion curve of each joint angle is shown in Fig. 3, which shows partial data from one subject walking on a flat. The features and trends of changes and peaks in each curve were evident. Consequently, the differences between these curves can be used to distinguish the five locomotion modes.

## Channel attention mechanism

The fundamental thought of the attention mechanism is to instruct the model to autonomously learn the most critical channel information within a specific task. Consequently, this approach significantly boosts the attention of the model on these features, ultimately yielding more efficient and precise classification or predictive outcomes. Notably, the channel attention mechanism (CAM) can further assist the model in focusing more on salient channels during feature processing, thereby enhancing the overall performance and generalization capability of the entire model. The distinctive mechanism holds immense application potential and latent value across multiple related fields, such as image processing, voice recognition, and natural language processing.

Generally, the archetypal example of CAM is squeeze-and-excitation networks (SENet). Figure 4 illustrates the fundamental structure of SENet, a model that compartmentalizes the CAM into squeeze and excitation stages. During this process, the primary function of squeeze lies in reducing global spatial information. Subsequently, features are learned

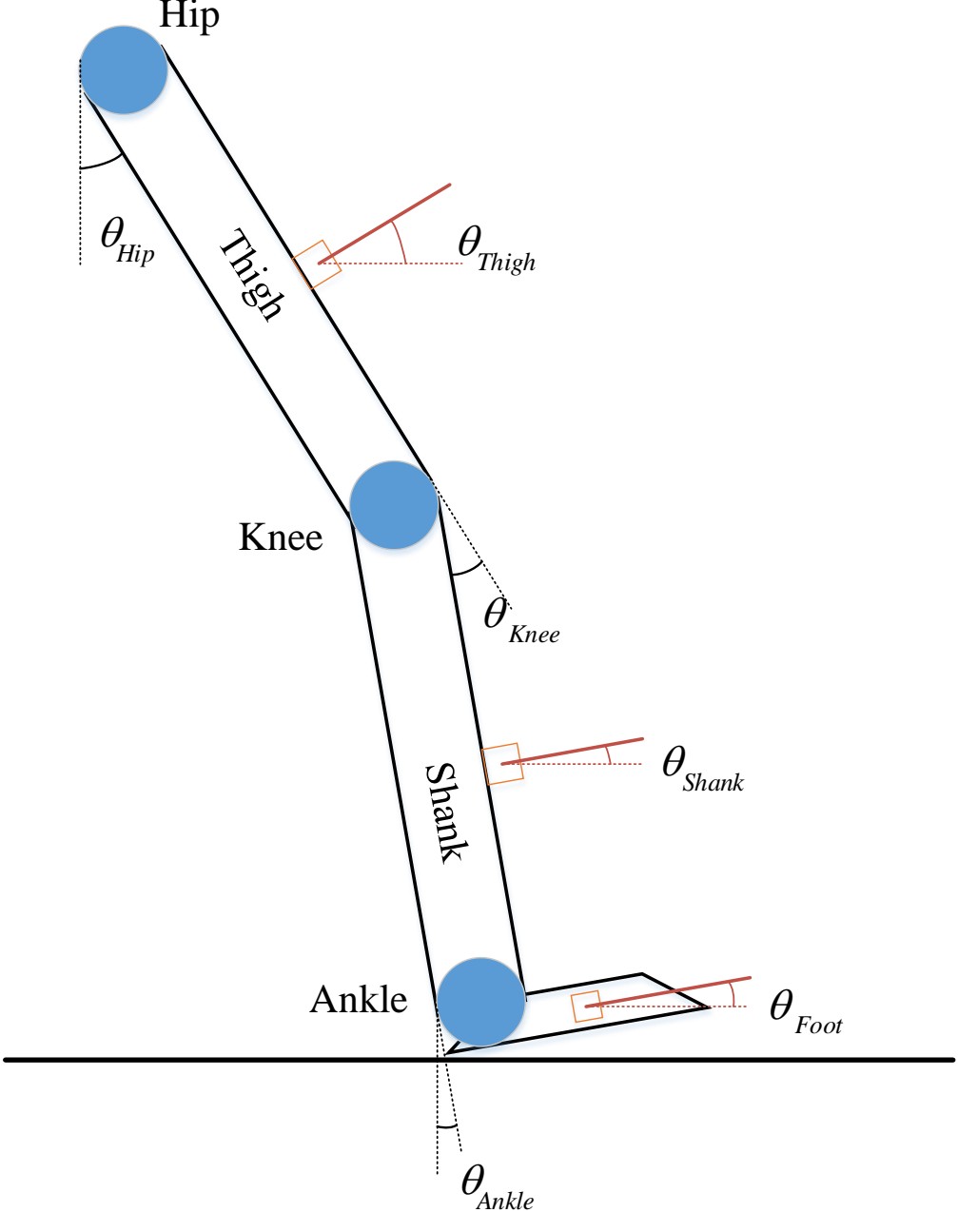

**Figure 2  Definition of joints and corresponding angles.** Take a single leg as an example.

across channels to ascertain the significance of individual channels. Eventually, through the excitation stage, distinct weights are assigned to individual channels. Within the diagram, $h$ and $w$ represent the dimensions of the feature matrix, while $c$ denotes the channel count.

Specifically, the SENet employs global average pooling to squeeze the feature map of each channel into a feature vector $[1,1,c]$. Subsequently, through a fully connected layer (FC1), the channel dimension of the feature map vector is reduced to $1/r$ of the original size,

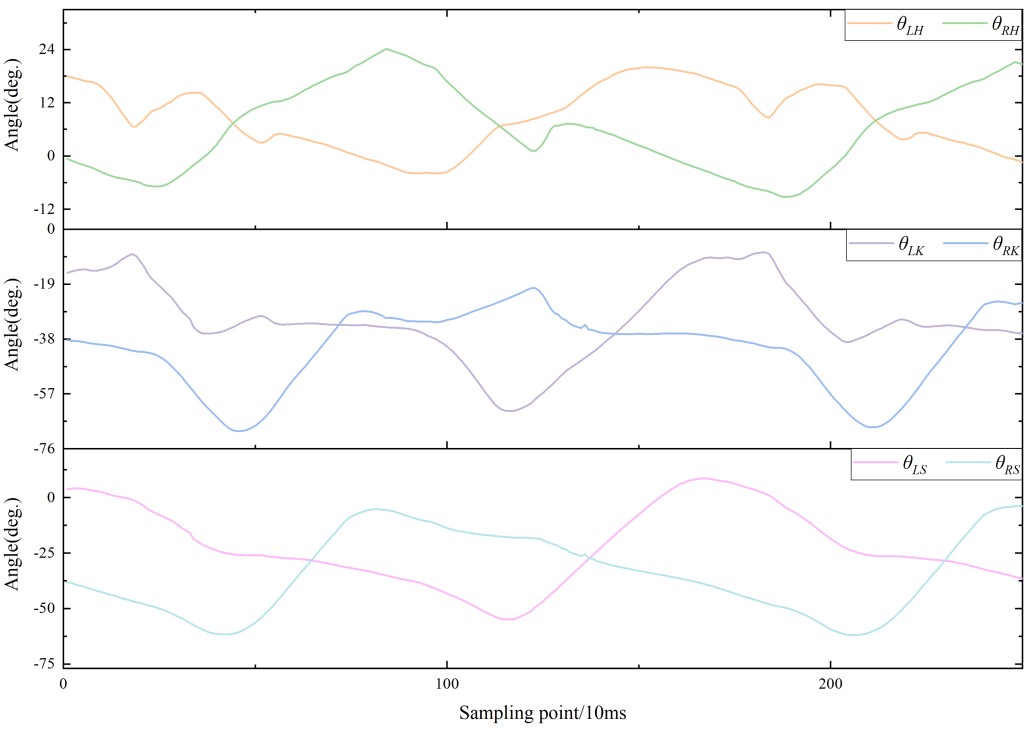

**Figure 3** Curves of walking on flat ground.

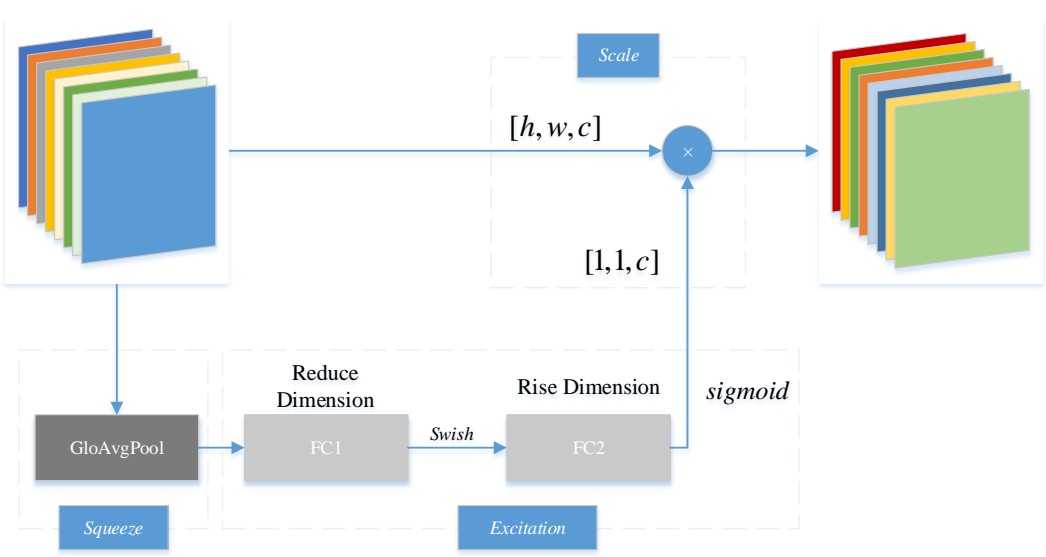

**Figure 4** Structure of SENet.

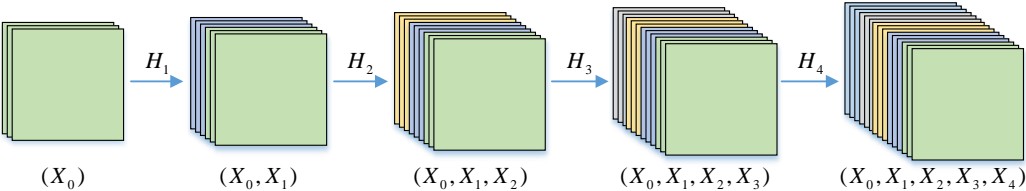

$(X_0)$     $(X_0, X_1)$     $(X_0, X_1, X_2)$     $(X_0, X_1, X_2, X_3)$     $(X_0, X_1, X_2, X_3, X_4)$

**Figure 5** **Connections of DenseNet network.** Squares represent the mapping of the input feature channels.

which is $[1,1,c/r]$; then, the feature undergoes a swish activation function; subsequently, it passes through another fully connected layer (FC2) to restore the feature map to the original $[1,1,c]$; thereafter, the feature undergoes the transformation of a sigmoid function into a normalized weight vector ranging from zero to one. Finally, the normalized weight is multiplied by the original input feature map channel-by-channel to generate the weighted feature map.

In essence, the SENet automatically learns feature weights in accordance with losses through a fully connected network. Instead of classifying based solely on the numerical value of each feature channel, the SENet enhances the weight of valuable channels. By acquiring the degree of importance for each channel in the feature map, criticality is then employed to assign weights to each feature. Subsequently, the neural network is focused on specific feature channels through the SENet. The primary objective is to amplify channels relevant to the current task while suppressing those feature channels that are irrelevant to the current task.

## Dense convolutional network

DenseNet (*Huang et al., 2017*) is a convolutional neural network architecture in which each layer takes the output of all previous layers as input. It can help the backward propagation of gradients during training and enables to train of deeper CNN. Feature multiplexing is a characteristic of DenseNet, which is achieved through connecting features on channels. For a neural network with $i$ layers, DenseNet contains $i(i+1)/2$ connections. It uses less calculation to achieve better performance. In DenseNet, all previous layers are connected as shown in Fig. 5. Meanwhile, the output at layer $i$ for the network is

$$X_i = H_i([X_0, X_1, \ldots, X_{i-1}]) \tag{3}$$

Where $i$ indicates the amount of network layers. $X_i$ indicates the output corresponding to the $i^{th}$ layer. $H_i$ is a series of combinations of non-linear transformations, including batch normalization, activation, convolution, and pooling.

Meanwhile, Dense Block, Transition Layer, and small growth rate are used in DenseNet. It makes the network narrower and reduces the calculation, effectively suppressing the problem of network overfitting. Traditional CNN networks typically rely on convolution and pooling to decrease the dimensionality of the feature map. In DenseNet, the layers are densely connected, necessitating the retention of the feature maps of the same dimensions

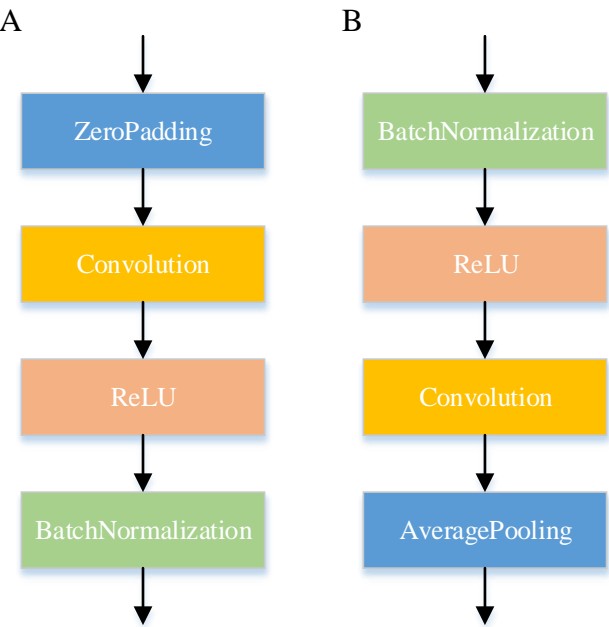

**Figure 6 Internal structure of the DenseNet network.** (A) is the Dense Block and (B) is the Transition Layer.

for successful forward propagation of the network. Thus, using the Dense Block and Transition Layer is an effective solution to this problem in DenseNet.

Figure 6 displays the internal of the DenseNet. In the Dense Block, the feature vectors are first zero-padded using the Zero-Padding layer to control the feature vector length and then passed by a Convolution layer. The computational efficiency of the network is optimized by reducing the dimensionality of feature maps in the Dense Block. Furthermore, the Transition Layer reduces the dimensions of feature maps while also decreasing the number of features.

## Long short-term memory network

LSTM, serving as a variant of recurrent neural networks (RNNs), possesses the ability to process input information through the analysis of time series. LSTM is used to tackle the problem of long-term dependency in conventional RNNs. Moreover, the problem of gradient loss or explosion can be tackled through the incorporation of LSTM within the network. The architecture of the LSTM is shown in Fig. 7.

The rounded rectangular box is a Memory Block, which mainly consists of three gating units and a memory unit. The three gating units comprise the forgetting, input, and output gates. The gating units filter and screen the valid information before the sequence data to the next Memory Block. The memory unit is primarily designed for information with long-term dependencies. Specifically, the calculations for each step are as follows:

$$f_t = \sigma \left( W_f \cdot [h_{t-1}, x_t] + b_f \right) \tag{4}$$

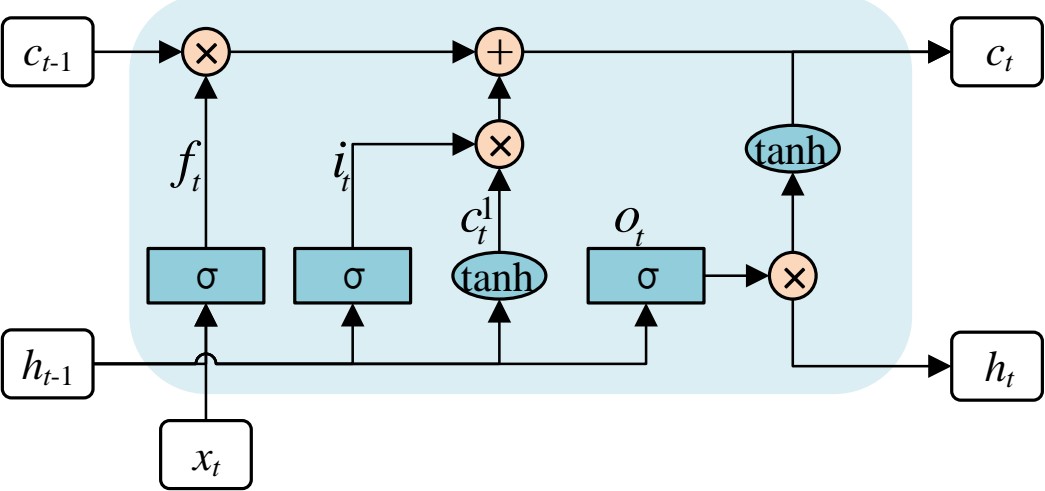

**Figure 7** **Internal structure of the LSTM network.**

$$i_t = \sigma(W_i \cdot [h_{t-1}, x_t] + b_i) \tag{5}$$

$$c_t^1 = \tanh(W_c \cdot [h_{t-1}, x_t] + b_c) \tag{6}$$

$$c_t = (f_t \times c_{t-1}) + (i_t \times c_t^1) \tag{7}$$

$$O_t = \sigma(W_o \cdot [h_{t-1}, x_t] + b_o) \tag{8}$$

$$h_t = o_t \times \tanh(C_t). \tag{9}$$

Where $x$ represents the feature vectors for the input; $h$ represents the feature vectors for the output; $c$ represents the input vector for cell activation, which is the core of the LSTM; $f$, $i$, $c^1$ and $o$ represent the output for each gating cell. Equation (4) is the first step of the LSTM network, called the forget gate layer. $\sigma$ is an activation function called sigmoid, which returns an output value within the range of zero to one. Meanwhile, some of these values are selectively ignored by the LSTM. The second step is Eqs. (5) and (6), called the input layer and tanh activation layer. The results obtained from the two additional layers are fed into the cell state. Eqs. (7) and (8) are the third and last steps, and the final output is shown in Eq. (9).

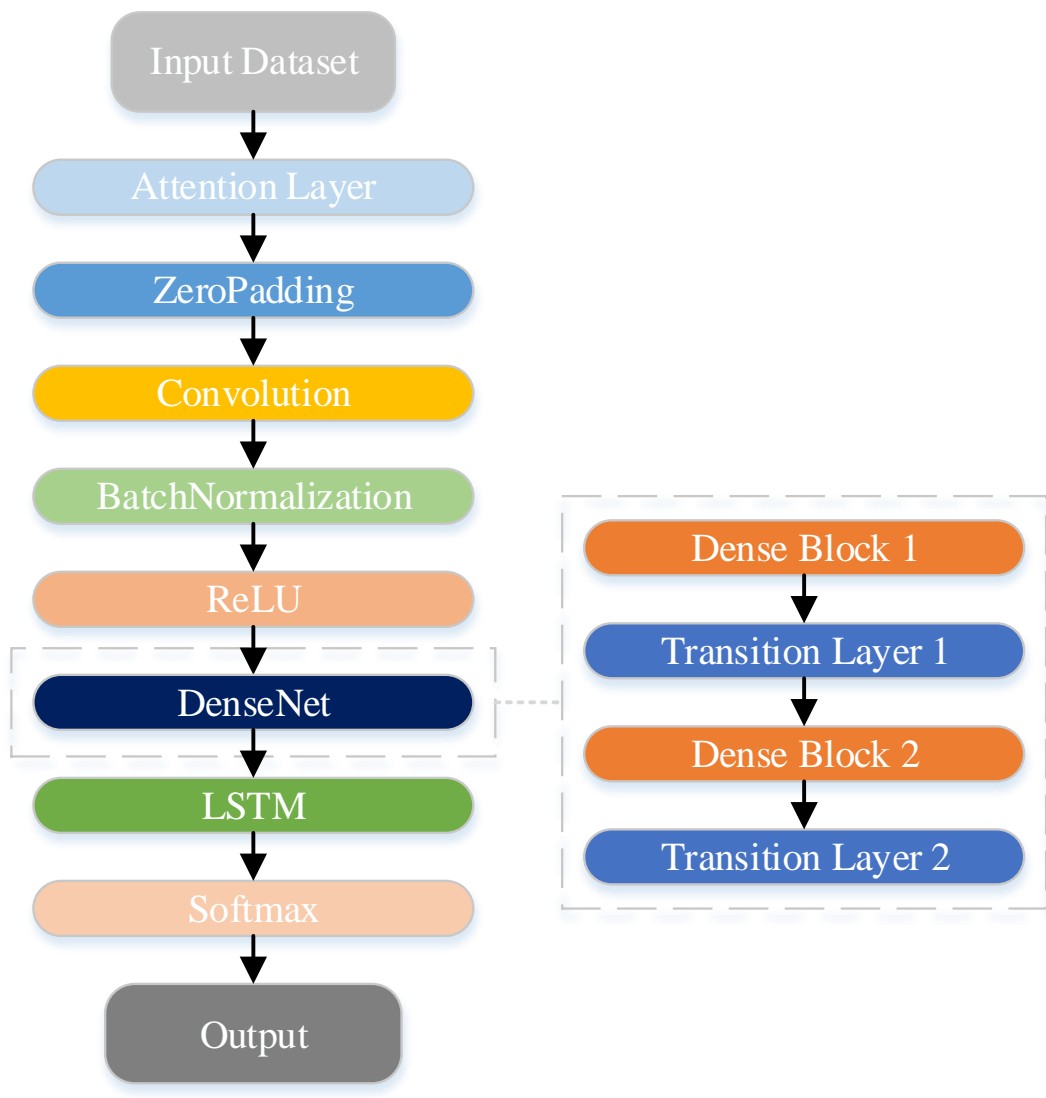

**Figure 8** Structure of the DenseNet-LSTM with the attention module.

### Hybrid SE-DenseNet-LSTM model

In this study, a hybrid DenseNet-LSTM model with an attention module is proposed. The DenseNet network yields feature mappings into the LSTM to reflect the sequence information of the features. Furthermore, the SENet module is introduced to enhance adaptation to the complex associations between channels, thereby improving model performance. Figure 8 illustrates the architecture of the DenseNet-LSTM with the attention module.

To avoid overfitting the model, sufficient data samples were generally expected when using deep neural networks. Meanwhile, the count of the network layers is correspondingly increased to enhance overall network performance. DenseNet is a model that effectively

**Table 1  Architecture of the proposed hybrid SE-DenseNet-LSTM model.**

| Layer | Feature map size | Configuration |
|---|---|---|
| Input | $1 \times 6 \times 8$ | – |
| Attention | $1 \times 6 \times 8$ | 8 |
| Convolution | $1 \times 10 \times 2$ | filters 2, $[3 \times 3 \; conv\,]$ |
| Dense Block 1 | $1 \times 10 \times 160$ | filters 64, $[3 \times 3 \; conv\,] \times 4$ |
| Transition | $1 \times 10 \times 256$ | filters 256, $[1 \times 1 \; conv\,]$ |
| Block 1 | $1 \times 5 \times 256$ | $2 \times 2$ average pooling, stride 2 |
| Dense Block 2 | $1 \times 5 \times 224$ | filters 128, $[3 \times 3 \; conv\,] \times 4$ |
| Transition | $1 \times 5 \times 512$ | filters 512, $[1 \times 1 \; conv\,]$ |
| Block 2 | $1 \times 2 \times 512$ | $2 \times 2$ average pooling, stride 2 |
| LSTM | $1 \times 64$ | – |
| Softmax | $1 \times 5$ | – |

mitigates the problem of vanishing gradients during training, while simultaneously minimizing the number of network parameters.

Table 1 shows the specifics of the network. The dataset comprises processed data derived from the original IMU signals, including the angles of the hip and knee joints, and the IMU signals obtained from the shank. The data is first passed through the SENet module and zero-padding into the convolutional layer. Next, all dense blocks possess an equal number of layers. Furthermore, a $3 \times 3$ convolutional layer is utilized to execute zero-padding, ensuring the size of feature maps remains constant. Correspondingly, a transition layer is employed after each dense block. Transition layers decrease the dimension of the feature map by utilizing a convolutional layer with a $1 \times 1$ kernel followed by a layer of average pooling. Lastly, the network output of DenseNet is transformed into a data format compatible with LSTM and subsequently inputted into LSTM. The ultimate classification result is garnered through the Softmax layer.

## Performance of evaluation

To evaluate the performance of the proposed model, four metrics are used as recall (Rec), precision (Pre), F1 score (F1), and accuracy (Acc) and represented as

$$Rec = \frac{TP}{TP + FN} \tag{10}$$

$$Pre = \frac{TP}{TP + FP} \tag{11}$$

$$F1 = 2 \cdot \frac{Pre \cdot Rec}{Pre + Rec} \tag{12}$$

$$Acc = \frac{TP + TN}{TP + FP + TN + FN}. \tag{13}$$

Where TP is the number of correct samples recognized as positive. FP is the number of false samples recognized as negative. TN is the number of correct samples recognized as negative. FN is the number of false samples recognized as wrong. Rec is the ratio of correctly predicted positive observations to the total actual positive observations. Pre is the ratio of correctly predicted positive observations to the total predicted positive observations. F1 is the weighted average of Pre and Rec. Acc is the ratio of correctly predicted observations to the total number of observations. Moreover, Eq. (14) is a confusion matrix to quantify and analyze the error distributions in the proposed model for locomotion mode recognition.

$$C = \begin{bmatrix} c_{11} & \cdots & c_{1j} \\ \vdots & \ddots & \vdots \\ c_{i1} & \cdots & c_{ij} \end{bmatrix}. \tag{14}$$

Where $c_{ij}$ represents the count of samples in which the $j^{th}$ locomotion mode is recognized as the $i^{th}$ locomotion mode. The diagonal elements show the quantity of correctly identified samples for the current locomotion mode. The remaining elements show the quantity of falsely identified samples.

Furthermore, to evaluate the effectiveness of the proposed model, the dataset was divided into two subsets with a ratio of 7:3, including training and testing. Based on the evaluation results obtained from the testing subset, the parameters of the model in this article were adjusted to enhance its overall ability of generalization.

# EXPERIMENTAL RESULTS AND DISCUSSIONS

## Preparations for the experiment

The experimental environment and the selection of hyperparameters for the SE-DenseNet-LSTM network will first be described. Experiments were conducted with Python version 3.8, Keras version 2.9.0, and TensorFlow version 2.9.1. Figure 9 shows the sliding window applied to the dataset in the experiment. An optimal sliding window can increase the utilization of the dataset and concurrently enhance the overall performance of the neural network. Therefore, the initial experiment was to determine the optimal sliding window size, utilizing Acc and the recognition rate of each locomotion mode as evaluated metrics. In which, the sliding window was set to stride by one sampling interval. In addition, the motion data of 10 subjects was used as a dataset for parameter selection of the model.

The impact of the sliding window on model performance is shown in Fig. 10. The proposed model was evaluated by various sizes of the sliding window, ranging from $2^n$ (where $n = 0, 1, \ldots, 4$). The results showed that FW is perfectly recognized for almost all sizes of the sliding window. However, once the sliding window size reached more than four, the Acc of the model was more than 97%. Conversely, Acc fell less than 96% for smaller window sizes. Based on the recognition rates of each locomotion mode, the optimal size was determined to be eight. Although Acc was slightly higher for window sizes of four and 16, they contained more outliers. Especially when the size was four, it showed a greater

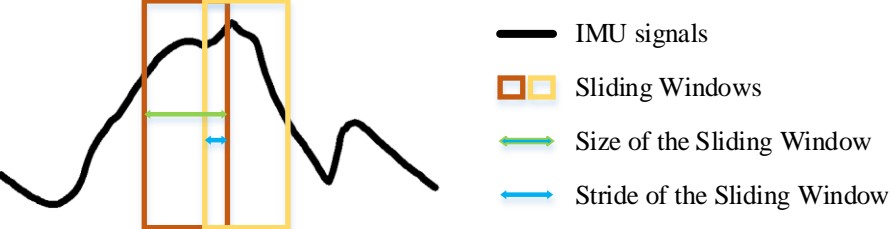

**Figure 9** Structure of the sliding window.

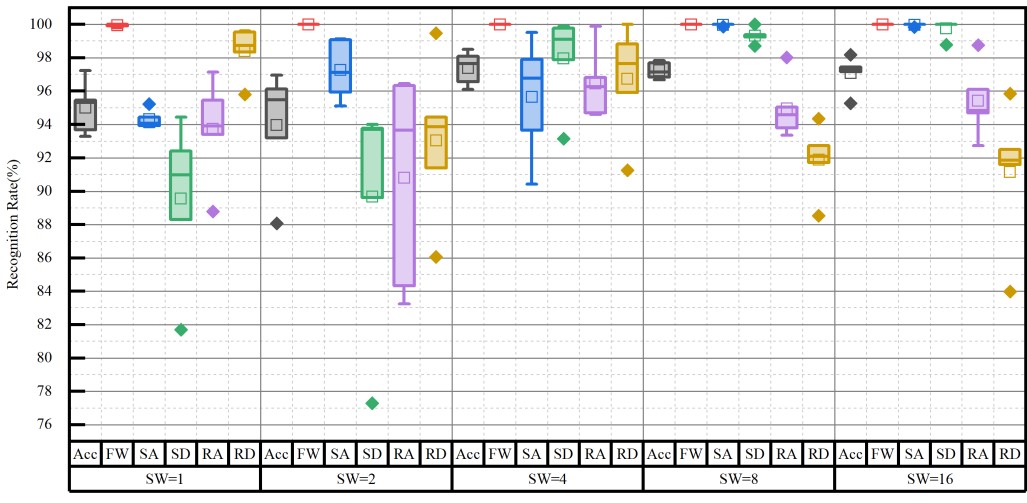

**Figure 10** Impact of the sliding window (SW) on network performance.

standard deviation in the recognition of each locomotion mode. As a result, the sliding window size of eight was chosen for subsequent experiments.

Hyperparameters play a crucial role in deep learning networks, including the number of neural units and layers. A suitable number can enhance the fitting ability of the network. On the contrary, inappropriate ones can reveal potential drawbacks, such as greater complexity, slower training times, and a higher probability of overfitting. Hyperparameters must be manually selected while constructing a neural network. Through experience and continuous experimentation, the best hyperparameters can be selected to optimize the performance of the network. Thus, the three major hyperparameters were experimentally selected and determined, including the learning rate, batch size, and size of hidden layers. Moreover, the activation function (ReLU) and optimizer (Adam) were also selected.

The learning rate (LR) determines the stride for updating the weights of the network. Reducing LR can lead to slower convergence of the model. In contrast, it may also lead to unstable convergence, causing the model to oscillate around the optimal solution. LRs were set at 1*e*-3, 1*e*-4, and 1*e*-5 for the experiments, and the impact of LR on network

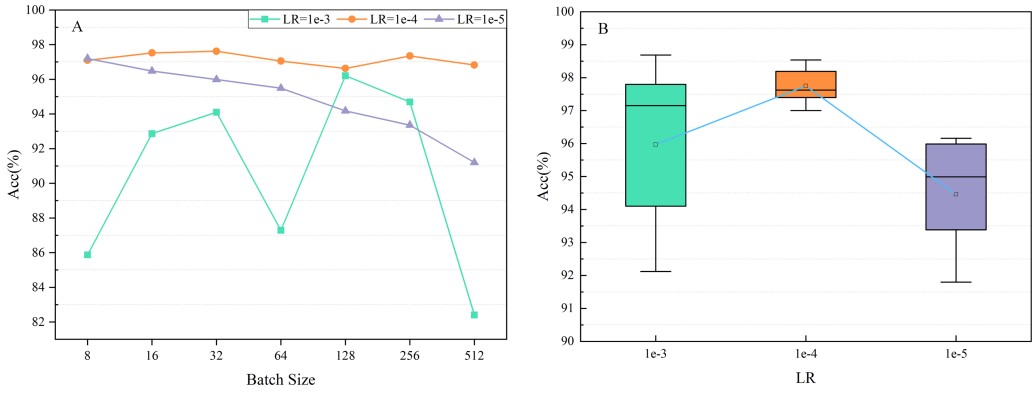

**Figure 11 Impact of the learning rate (LR) on network performance.**

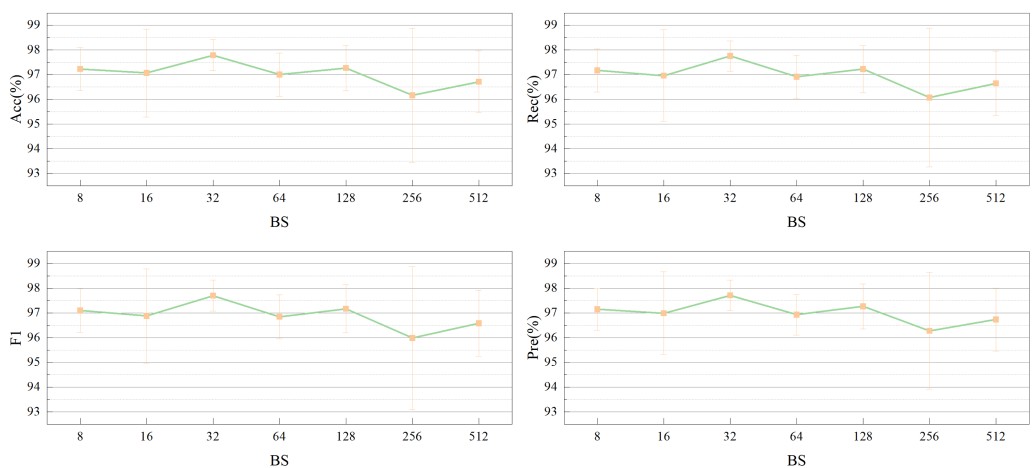

**Figure 12 Impact of the batch size (BS) on network performance.**

performance is shown in Fig. 11A. The results indicated that the Acc of the proposed model remained around 98% when selecting a varying batch size and LR of 1$e$-4.

Furthermore, several experiments were carried out with different random seeds for each LR to verify that LR of 1$e$-4 was the best option. Figure 11B shows the mean accuracy (0.9775) and standard deviation (0.00616) achieved when adjusting the LR to 1$e$-4, with the best performance of all experiments. Therefore, LR was set at 1$e$-4 for all experiments.

Similarly, Fig. 12 depicts an experimental exploration of batch size selection at a LR of 1$e$-4. The performance of the model in this article was compared for different batch sizes of $2^n$ (where $n = 3, 4, \ldots, 9$) with Acc, Rec, F1, and Pre as evaluation metrics. The highest performance was achieved with each evaluation metric of more than 0.977 at a batch size of 32.

The final selection was made for the size of hidden layers (HL) in the LSTM. The performance of the model can be also significantly influenced by the size of HL. To tackle long-term dependence on time series, it is more effective to increase the size of HL

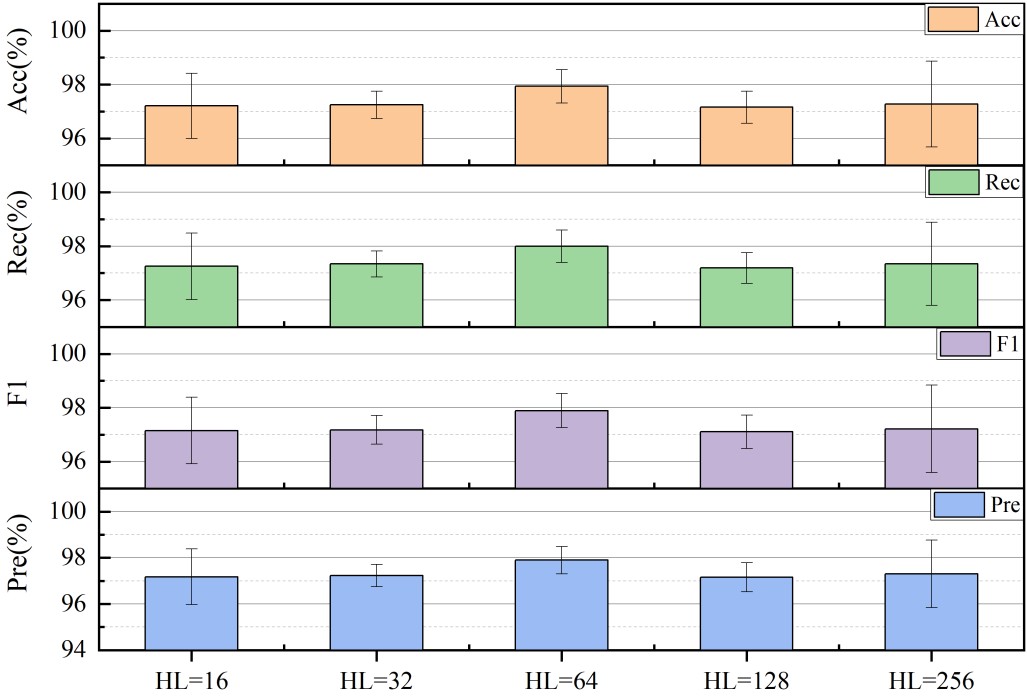

**Figure 13** Impact of the hidden layer (HL) size on network performance.

**Table 2** Confusion matrix of DenseNet-LSTM for one subject (%).

| Locomotion Mode | | Predicted Label | | | | |
|---|---|---|---|---|---|---|
| | | FW | SA | SD | RA | RD |
| | FW | 100.00 | 0.00 | 0.00 | 0.00 | 0.00 |
| | SA | 0.00 | 100.00 | 0.00 | 0.00 | 0.00 |
| True Label | SD | 0.00 | 0.00 | 97.19 | 0.00 | 2.81 |
| | RA | 1.12 | 0.70 | 1.09 | 97.09 | 0.00 |
| | RD | 0.97 | 0.24 | 0.46 | 0.00 | 98.33 |

appropriately, which can improve the recognition rate and generalizability of the network. However, excessively large HL can elevate the complexity of the network, resulting in extended computational periods. Experiments were conducted on the network by adjusting the HL size to $2^n$ (where $n = 4, 5, \ldots, 8$), using Acc, Rec, F1, and Pre as evaluation metrics. The model showed optimal performance when HL had a size of 64 in Fig. 13.

## Experimental results of single locomotion mode recognition

Experimental results for one subject were analyzed after determining the model hyperparameters. Table 2 shows recognition results for one subject in five distinct locomotion modes using the proposed hybrid SE-DenseNet-LSTM model. The diagonal line of the table showed the recognition rate of locomotion modes for the subject, with an overall mean recognition rate of 98.53%.

**Table 3  Results of ablation experiments.**

| Model | DenseNet | LSTM | Attention | Acc(%) | Rec(%) | F1 | Pre(%) |
|---|---|---|---|---|---|---|---|
| A | ✓ | – | – | 96.01 | 96.12 | 95.92 | 96.10 |
| B | – | ✓ | ✓ | 97.03 | 97.04 | 96.96 | 97.09 |
| C | ✓ | ✓ | – | 96.35 | 96.42 | 96.26 | 96.44 |
| This article | ✓ | ✓ | ✓ | 97.93 | 98.00 | 97.89 | 97.90 |

**Table 4  Comparisons of the mean accuracy of the four models (%).**

| Model | Locomotion Mode | | | | |
|---|---|---|---|---|---|
| | FW | SA | SD | RA | RD |
| A | 100.00 | 100.00 | 97.00 | 93.87 | 89.62 |
| B | 99.93 | 99.97 | 98.30 | 92.78 | 94.48 |
| C | 100.00 | 100.00 | 97.47 | 93.96 | 90.77 |
| This article | 100.00 | 100.00 | 99.36 | 96.05 | 94.09 |

Four models (A, B, C, and this article) using the proposed model were individually created to perform ablation experiments. In order to confirm the superiority of the model in this article, the performance of each model was systematically compared through various evaluation metrics. For fairness, the hyperparameters identified in the previous section were utilized across all models. Mean Acc, Rec, F1, and Pre for the four models are shown in Table 3. The proposed model demonstrated a performance improvement of about 2% in mean Acc compared to the traditional DenseNet network.

Table 4 shows the mean accuracy of various locomotion modes for these four models. All models were almost recognized with perfect accuracy for FW and SA, and the mean recognition rate for SD exceeded 97%. DenseNet has been shown to enhance the transmission of features to some extent. The recognition rates of FW and SA of model B were slightly reduced. However, recognition rates for both SD and RD improved to varying degrees. The addition of the SENet module effectively addressed the effect of the small ramp on locomotion mode recognition. The recognition rate of various locomotion modes combined with the LSTM layer was substantially the same as model A. Nevertheless, model C improved the recognition rate for RD by 1% and effectively dealt with long-term temporal dependencies. Compared to models A, B, and C, the proposed model, DenseNet-LSTM with attention module, combines the advantages of DenseNet and LSTM. It has an efficient feature extractor and a good capacity to tackle long-term dependency problems. The SENet module assists the proposed model in prioritizing the most relevant information at any given time by adjusting weights. In the proposed model, the locomotion modes of FW, SA, and SD were perfectly recognized. Specifically, the recognition rates for both RA and RD had significantly improved. Therefore, the proposed model has a higher accuracy.

## Online recognition

In order to investigate the practical usability and efficacy of the proposed SE-DenseNet-LSTM model in dynamic environments, an experiment on online recognition was

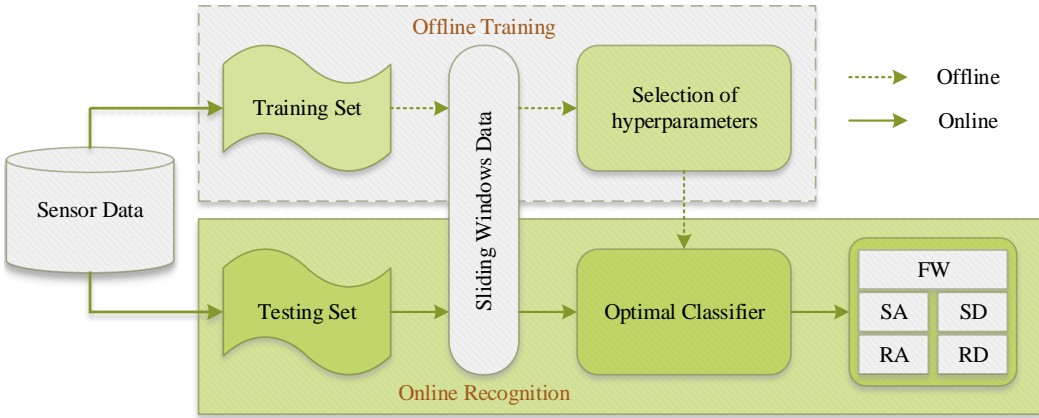

**Figure 14** The procedure for the online testing with data from five untrained subjects.

**Table 5** Results of online recognition with data from five untrained subjects.

| Subjects | Acc(%) | Rec(%) | F1 | Pre(%) |
|---|---|---|---|---|
| S1 | 95.27 ± 2.47 | 95.16 ± 2.52 | 95.25 ± 2.46 | 95.46 ± 2.31 |
| S2 | 93.59 ± 3.98 | 93.47 ± 4.22 | 93.54 ± 4.10 | 93.94 ± 3.53 |
| S3 | 94.00 ± 2.15 | 93.99 ± 2.15 | 94.20 ± 2.11 | 94.66 ± 2.04 |
| S4 | 96.99 ± 1.63 | 97.11 ± 1.54 | 96.99 ± 1.59 | 97.14 ± 1.39 |
| S5 | 92.35 ± 2.09 | 92.65 ± 1.98 | 92.25 ± 2.06 | 92.59 ± 1.80 |
| Mean | 94.44 ± 2.46 | 94.47 ± 2.48 | 94.44 ± 2.46 | 94.76 ± 2.21 |

conducted. Simultaneously, to extend the investigation to subjects with diverse physical features, the already-trained model was utilized for the validation of data from five additional untrained subjects. The procedure for the online testing with data from five untrained subjects is illustrated in Fig. 14.

Specifically, out of 15 subjects, the training set encompassed the motion data of 10 subjects, while the testing set encompassed the motion data of the remaining five subjects. In particular, in the process of offline training, the training set underwent processing using a sliding window technique. An optimal SE-DenseNet-LSTM classifier was obtained after selecting suitable hyperparameters. To uphold experimental consistency, the optimal classifier from the offline training was employed to validate the testing set in online recognition, thereby achieving recognition of five distinct locomotion modes. The overall recognition results are illustrated in Table 5.

On the one hand, by observing the results of the different evaluation metrics, the mean value among the five subjects in each metric reached 94.44%, while the average standard deviation was maintained at about 2.46%. The results signified that the SE-DenseNet-LSTM model demonstrated superior overall accuracy in recognizing the movement data of different subjects, capable of effectively identifying distinctive movements. On the other hand, particularly in relation to individual subjects, the recognition rate of S4

was the highest. Concretely, the average accuracy was as high as 97%, while the average standard deviation was also hovered around only 1.5%. The results were comparable to the recognition results of offline training. Regrettably, however, S5 performed somewhat short, with all indicators reaching merely 92%.

Overall, certain differences were observed in Acc, Rec, F1, and Pre amongst the different subjects. The differences may be a consequence of variations in the movement characteristics, walking postures, and physical conditions of diverse subjects. In essence, due to the untrained movement data of these five subjects, it is logical that the online recognition rate is lower relative to offline training. Nonetheless, the proposed SE-DenseNet-LSTM model still exhibited a high recognition level of locomotion mode when applied to untrained subjects.

## The impact of ramp data on recognition results

As previously mentioned, the proposed model has effectively enhanced the accuracy of RA and RD. In the recent study (*Cheng, Bolívar-Nieto & Gregg, 2021*), the ramp data was categorized as FW since the maximum angle of the ramp terrain used for the experiment was only 10°. This approach can improve the overall recognition rate for four locomotion modes, including sitting, FW, SA, and SD. However, during the recognition process, due to the FW being incorrectly classified as other locomotion modes from ramp data, it decreases the recognition rate of the locomotion mode of FW. Therefore, in this section, RA and RD are also categorized as FW to further validate the effect of ramp data on recognition results.

Figures 15, 16 and 17 depict the angle curves of all input data for one subject in five terrains, and the included angles are shown in Eq. (1). In which, the $x$-axis and $y$-axis represents the number of samples and the angle of joints, and the dashed pink line indicates one sampling period of FW. For comparison, the motion angle curves of joints for one gait cycle are shown in each subplot, with the motion data for the remaining four terrains being compared with FW. As can be seen in Figs. 15A, 16A, and 17A, the trends of the angular curves for RA and RD are essentially the same as those for FW, with only slight differences. On the contrary, in Figs. 15B, 16B and 17B, the data for SA and SD differed significantly from those for FW. In particular, changes in the angles of hip and knee joints are much greater, which is consistent with the actual situation of walking. Additionally, for the comparison of the gait cycle, walking on stairs is about 20 more sampling points than walking on flat ground or ramps. Compared with SA and SD, the data characteristics of RA and RD are highly similar to FW, which may increase the misclassification of these three locomotion modes.

To evaluate the effect of ramp data on recognition results, RA and RD data were merged into FW, and the proposed model was trained and tested in three locomotion modes, repeating the process five times. The recognition results were compared with the original five locomotion modes. Confusion matrixes in Tables 6 and 7 recorded the recognition results of each locomotion mode, including the mean recognition rate (mean) and standard deviation (MSE). Table 6 shows the confusion matrix after mixing the flat and ramp data, with recognition rates for both SA and SD reaching over 99%. However, the recognition rate of FW was only 95.775% ± 2.595 with more misclassifications. In Table 7, by separating

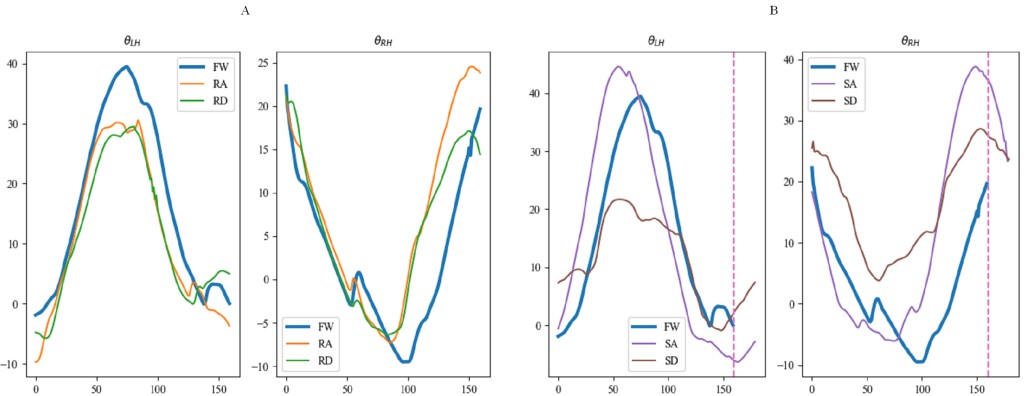

**Figure 15    The angular curves of the hip for one subject in five terrains.**

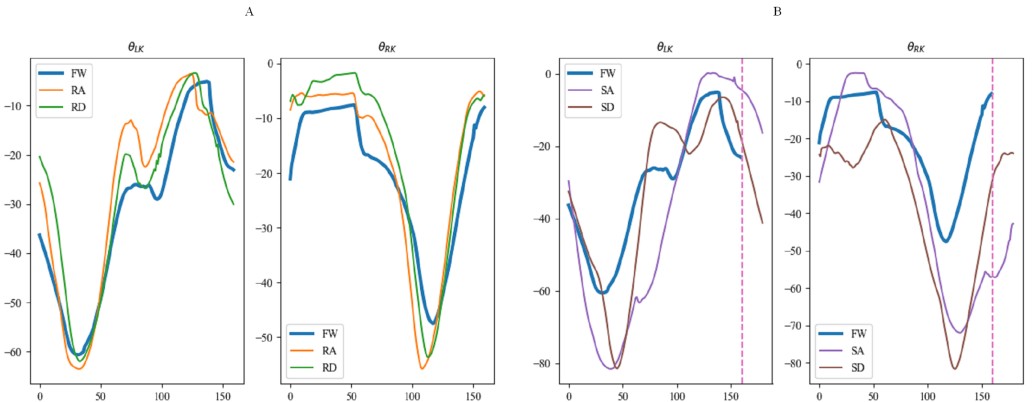

**Figure 16    The angular curves of the knee for one subject in five terrains.**

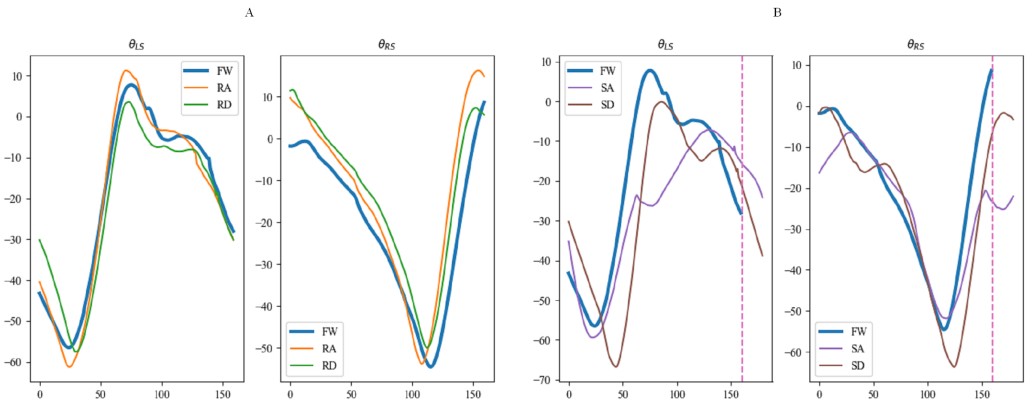

**Figure 17    The angular curves of the shank for one subject in five terrains.**

**Table 6  Confusion matrix (mean ± MSE) after mixing flat and ramp (%).**

| Locomotion Mode | | Predicted Label | | |
|---|---|---|---|---|
| | | FW(RA/RD) | SA | SD |
| True Label | FW(RA/RD) | 95.76 ± 2.595 | 2.42 ± 2.498 | 1.83 ± 1.633 |
| | SA | 0.03 ± 0.033 | 99.89 ± 0.116 | 0.08 ± 0.083 |
| | SD | 0.07 ± 0.101 | 0.68 ± 0.406 | 99.25 ± 0.472 |

**Table 7  Confusion matrix (mean ± MSE) after separating flat ground and ramp (%).**

| Locomotion Mode | | Predicted Label | | | | |
|---|---|---|---|---|---|---|
| | | FW | SA | SD | RA | RD |
| True Label | FW | 100.00 ± 0.000 | 0.00 ± 0.000 | 0.00 ± 0.000 | 0.00 ± 0.000 | 0.00 ± 0.000 |
| | SA | 0.00 ± 0.000 | 100.00 ± 0.000 | 0.00 ± 0.000 | 0.00 ± 0.000 | 0.00 ± 0.000 |
| | SD | 0.00 ± 0.000 | 0.00 ± 0.000 | 99.36 ± 0.628 | 0.05 ± 0.087 | 0.58 ± 0.550 |
| | RA | 1.63 ± 0.994 | 0.8 ± 0.562 | 1.42 ± 0.896 | 96.05 ± 2.441 | 0.07 ± 0.064 |
| | RD | 2.76 ± 0.556 | 1.23 ± 0.473 | 1.25 ± 0.415 | 0.66 ± 0.555 | 94.09 ± 1.376 |

the flat and ramp data, FW was perfectly recognized with a significantly higher recognition rate. Moreover, the recognition rates for SA and SD were comparable to those in Table 6. RA and RD exhibited relatively lower recognition rates, yet still exceeding 92%.

Figure 18 shows the comparison of recognition results before (non-mixing data) and after (mixing data) mixing RA and RD data into FW, using various evaluation metrics. In the non-mixing data, Acc, Rec, and F1 were superior to the mixing data, whereas Pre was the opposite. This was due to the uneven distribution of sample categories after mixing ramp and flat data. Therefore, in the mixing data, the model focused more on accurately predicting minority categories, thus obtaining a higher Pre. In the non-mixing data, the model emphasized more overall accuracy, hence the other three metrics were higher. In conclusion, the proposed model can still effectively recognize the three locomotion modes FW, RA, and RD, even in situations where ramp data is not obvious. Although ramp data has some influence on recognition results, the overall recognition rate can reach 98% for the proposed model.

## Compare with other methods

A hybrid DenseNet-LSTM model with an attention module is proposed. IMU sensors were utilized to capture information on human posture, while pressure sensors were utilized for the data collection on plantar pressure. Moreover, six original features were extracted into the neural network, and recognition of five locomotion modes was achieved. Consequently, the proposed model combined the strengths of DenseNet and LSTM to facilitate robust feature extraction from the data. The proposed model further enhanced the recognition rate of the locomotion mode, even when faced with a relatively small degree of the ramp.

Furthermore, training time and testing time were defined, representing the duration from start to end of each step in the training and testing of a neural network. Experimental results showed the proposed hybrid model achieved a training time of 66 ms and a testing

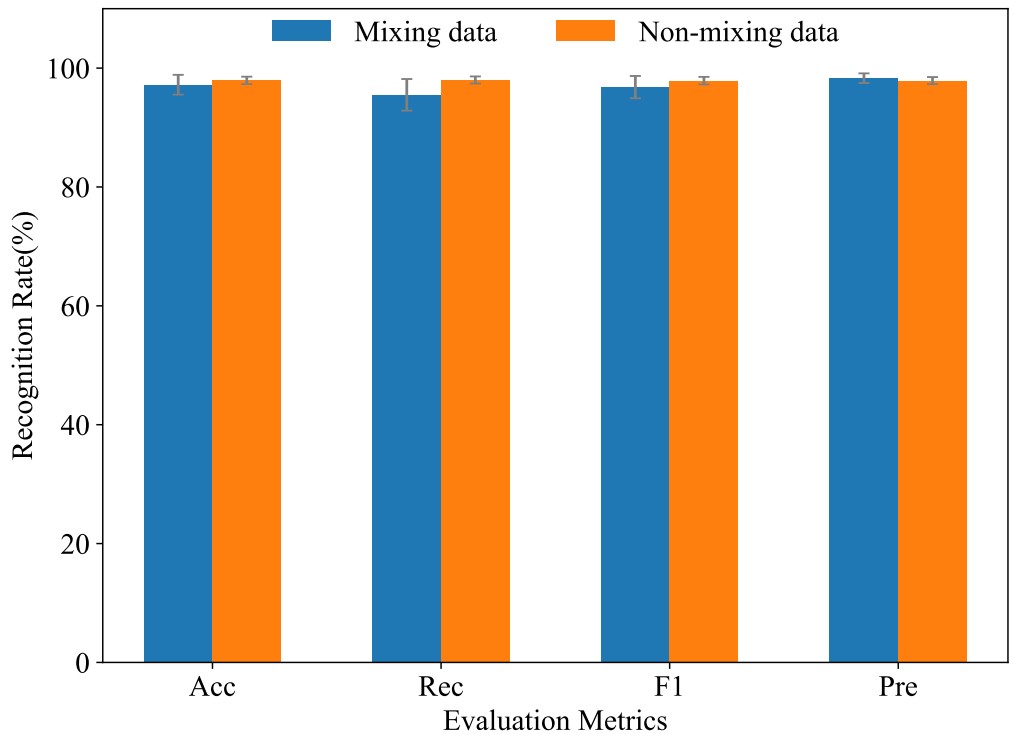

**Figure 18** **Comparisons of mixing and non-mixing ramp data on recognition rates.**

time of 9.4 ms, which satisfied the 10 ms sampling interval need. To analyze the early swing of the lower limb in humans, *Su et al. (2019)* used the motion data from IMUs to serve as input for the CNN. They obtained 94.15% in the recognition rate of locomotion modes. *Chung et al. (2019)* developed an LSTM network to train multimodal physical sensor data with a recognition rate of 92.18%. Naturally, many other methods have been employed for locomotion mode recognition of exoskeleton robots, just as SVM (*Zheng et al., 2022b*), hidden Markov model (HMM) (*Liu et al., 2017*), dynamic time warping (DTW) (*Zheng et al., 2022a*), KNN (*Zhang et al., 2023*) and linear discriminant analysis (LDA) (*Young & Hargrove, 2015*). Table 8 shows the results of the comparison with other methods. In contrast, the proposed SE-DenseNet-LSTM had a higher recognition rate.

## The impact of noise on model performance

The proposed model has an improvement in recognition rates compared to known methods. Nevertheless, noise caused by various factors can affect recognition results in practical applications. Therefore, it is essential to quantify the effect of noise on the overall performance for the proposed model. Common Gaussian white noise was selected as the noise disturbance for the experimental data. The noise disturbance in the practical environment was simulated by Gaussian white noise with different signal-to-noise ratios. Acc was used as the performance metric of the model against interference. Figure 19 shows the trend of the hip angle on the right leg in FW for the mean $\mu = 0$ and different standard deviations ($\sigma = 0,1,5,10,13,15$).

**Table 8  Results of the comparison with other methods.**

| Source | Sensors | Feature extraction | Classifier | Position of sensors | Time (ms) Training/Testing | Acc (%) |
|---|---|---|---|---|---|---|
| *Zheng et al. (2022b)* | IMUs | Manual | SVM | Healthy side | – | 89.61 |
| *Zheng et al. (2022a)* | IMUs | Manual | DTW | Healthy side | – | 90 |
| *Young & Hargrove (2015)* | IMU, axial load cell, motor current | Manual | LDA | Prosthesis | – | 93.9 |
| *Liu et al. (2017)* | Accelerometer, gyroscope, pressure sensor | Manual | HMM | Prosthesis | – | 95.8 |
| *Chung et al. (2019)* | IMUs/ Magnetometer/ Gyroscope | Automatic | LSTM | Healthy side | – | 92.18/ 85.22/ 78.33 |
| *Su et al. (2019)* | IMUs | Automatic | CNN | Healthy side | – | 94.15 |
| *Zhang et al. (2023)* | IMU, Axial load cell, Angle encoder | Automatic | ImprovedKNN | Prosthesis | – | 96.66 |
| This article | IMUs | Automatic | DenseNet-LSTM | Healthy side | 66/9.4 | 97.90 |

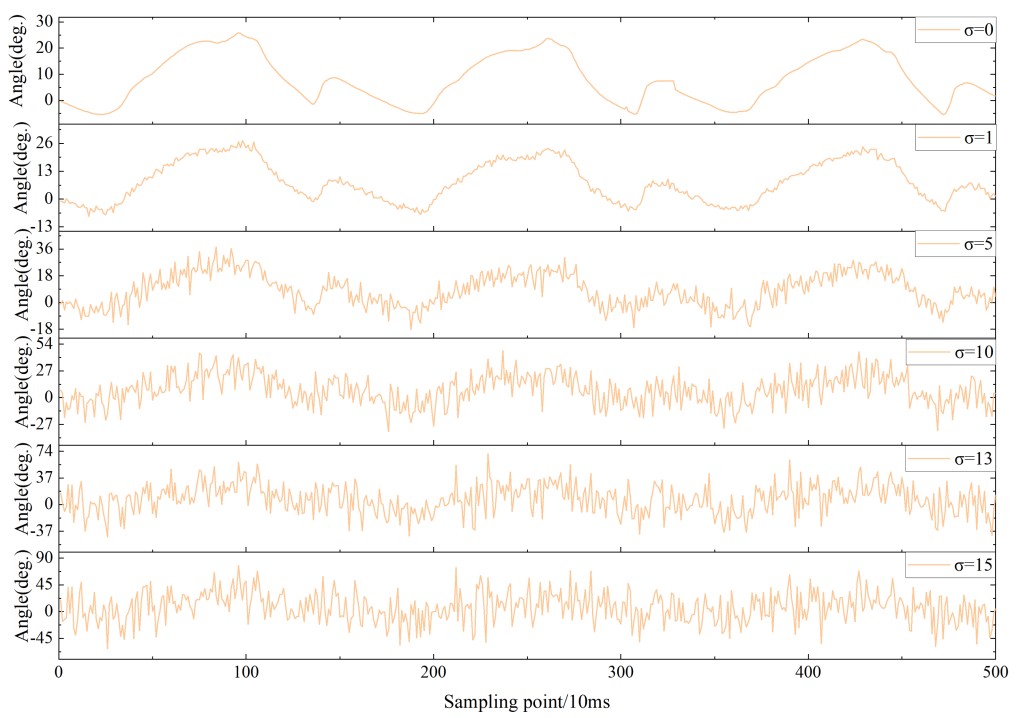

**Figure 19  The impact of varying degrees of white Gaussian noise on the right hip joint.**

The Acc trend curves of the model are shown for varying levels of $\sigma$ in Fig. 20. Acc of the model can still reach over 90% when $\sigma<5$, indicating a high resistance to noise. However,

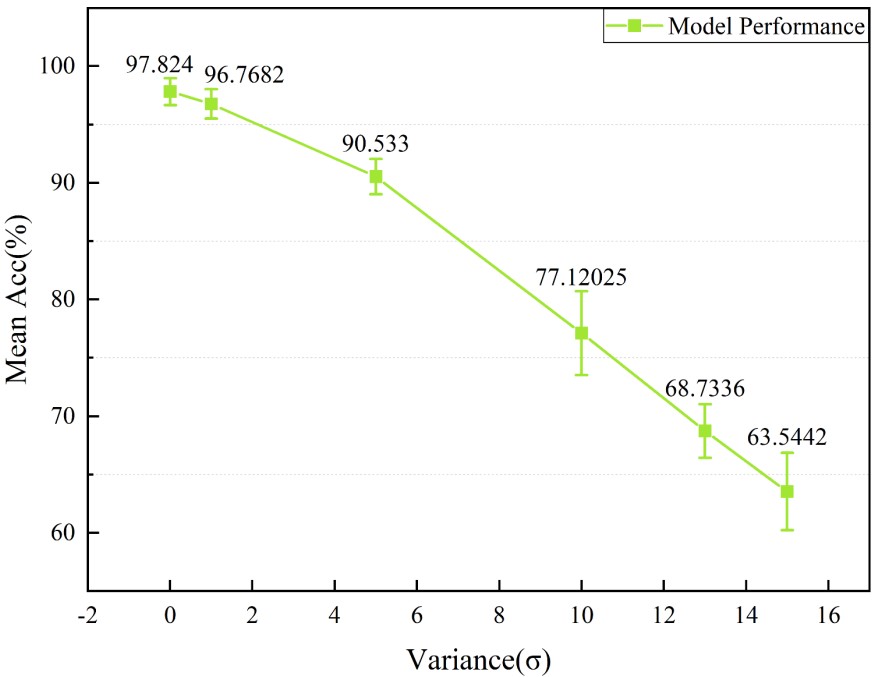

**Figure 20** **The impact of white Gaussian noise on the overall performance for the model.**

as the intensity of the noise gradually increases, especially after $\sigma \geq 10$, the Acc of the model significantly decreases. Overall, the proposed model exhibits some noise immunity.

## Lightweight model

DenseNet is fundamentally devised by establishing a network architecture *via* dense connectivity, enabling each layer to be directly linked with all previous layers. The structure accomplishes efficient transmission and recycling of features, thereby addressing the challenges of vanishing gradients and feature propagation prevalent in traditional CNNs. Nonetheless, various dense blocks and transition blocks are encompassed in the DenseNet. Each dense block is composed of multiple convolutional layers. The transition blocks, which control the dimensionality of feature maps, comprise pooling layers and convolutional layers. Consequently, the proposed model suffers from high parameter complexity and difficulty in training.

The computational load imposed by convolutional operations can be reduced by both grouped convolution and depthwise separable convolution (DSConv). However, grouped convolution merely replaces the convolutional operation into several groups, with no information interaction between them. Therefore, ordinary convolution is substituted for grouped convolution during the convolutional operation, the method may result in the loss of feature information. In contrast, DSConv is a technique that separates the ordinary convolutional process into two parts, depthwise convolution and pointwise convolution using different convolutional filters. The separation decouples the channel and spatial correlations of the convolution. Specifically, in the depthwise convolution, convolution

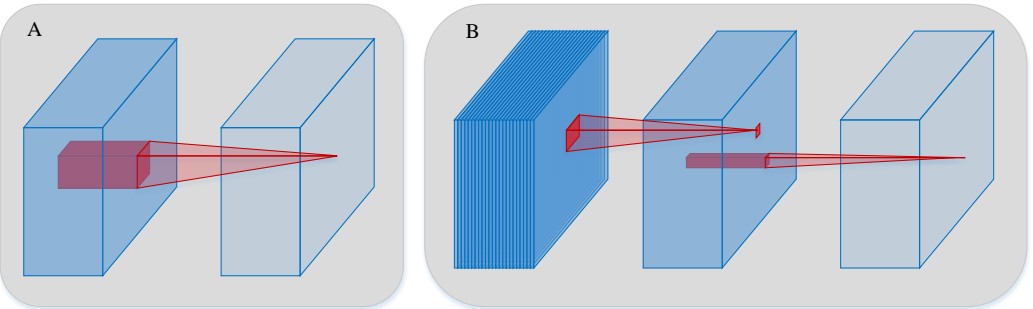

**Figure 21** **The processes of both conventional convolution and DSConv.** (A) is conventional convolution. (B) is DSConv.

**Table 9** **Results of replacing conventional convolution with DSConv.**

| Groups | Dense Block | Transition Block | Other Conv | Acc (%) | F1 (%) | Quantity of Parameters (M) | Size of Model (MB) |
|---|---|---|---|---|---|---|---|
| GP1 | ✓ | – | – | 96.02 ± 1.18 | 95.94 ± 1.21 | 4.59 | 14.09 |
| GP2 | – | ✓ | – | 96.83 ± 0.81 | 96.77 ± 0.83 | 7.91 | 24.01 |
| GP3 | – | – | ✓ | 96.81 ± 1.63 | 96.73 ± 1.66 | 7.90 | 23.99 |
| GP4 | ✓ | ✓ | – | 96.93 ± 2.00 | 96.86 ± 2.05 | 4.60 | 14.11 |
| GP5 | ✓ | – | ✓ | 96.20 ± 1.57 | 96.13 ± 1.60 | 4.59 | 14.09 |
| GP6 | – | ✓ | ✓ | 96.90 ± 1.24 | 96.83 ± 1.28 | 7.91 | 24.01 |
| GP7 | ✓ | ✓ | ✓ | 96.27 ± 1.65 | 96.23 ± 1.63 | 4.60 | 14.11 |
| This article | – | – | – | 97.93 ± 0.62 | 97.89 ± 0.63 | 7.90 | 23.99 |

computation independently is applied to each channel, thus dramatically reducing the computational load on the network. Pointwise convolution is utilized to fuse the channels to ensure the integrity of the feature information. Thus, without compromising the accuracy of the network model, both the computational load and the number of parameters in the convolutional operation can be effectively reduced in the DSConv. The processes of both conventional convolution and DSConv are illustrated in Fig. 21.

In this section, to mitigate the substantial parameter count introduced by feature reuse in the DenseNet, DSConvs are employed to replace the conventional convolutions in the proposed SE-DenseNet-LSTM model. To evaluate the effect of replacement by different DSConvs, the conventional convolutions are substituted in the dense block and transition block of the DenseNet, as well as the other ordinary convolutions employed during training, with DSConvs. The experimental results are depicted in Table 9.

Acc and F1 are selected as critical performance indicators, with both the number of parameters and the size of the proposed model serving as key evaluation metrics. The "✓" in the table signifies that common convolutions in the module have been replaced with DSConvs, while "-" denotes the opposite. A scrutinizing observation reveals that the size of the model and the number of parameters are substantially reduced by replacing ordinary convolutions within a dense block with DSConvs. Nevertheless, the Acc and F1 of the

model can still be maintained at 96% even as the number of parameters is decreased. In comparison with the proposed SE-DenseNet-LSTM model, the number of parameters has been virtually halved, but there is no discernible decline in the overall performance of the model. Thus, the application of deep separable convolutions to achieve the lightweight of the proposed model is successful.

## CONCLUSIONS

A hybrid model based on SE-DenseNet-LSTM is proposed for human locomotion mode recognition. Specifically, the DenseNet automatically extracts features from IMU data into LSTM to classify five locomotion modes of different terrains. To further enhance the model performance, the SENet module is incorporated to extract shallow features of the IMU data at a deeper level. Notably, the proposed model can maintain high recognition rates, even when the features of the ramp data are not immediately apparent. Overall, the proposed model has a high accuracy and robustness compared with other known algorithms.

In this article, only five locomotion modes have been recognized, yet the model has enormous potential for scalability, such as real-time recognition of human locomotion modes or more complex locomotion modes. In particular, the lower limb exoskeleton is useful both as a weight-bearing aid for healthy individuals and as a device to assist individuals with disabilities in walking. Furthermore, it is worthwhile to explore recognizing the transitions between different locomotion modes with greater precision. Therefore, the proposed model has limitations, and there are considerable areas for improvement in future work. The dataset can be expanded, for instance, to recognize locomotion modes for subjects with varying body weights, walking speeds, or weight loads to enhance the ability of generalization for the model.

## ACKNOWLEDGEMENTS

The authors would like to express gratitude to all participants.

### Funding

This research was funded by the Natural Science Foundation of Hubei Province (No. 2022CFA007) and the Hubei University of Technology Ph.D. Research Startup Fund Project (No. BSQD2020014). The funders had no role in study design, data collection and analysis, decision to publish, or preparation of the manuscript.

### Grant Disclosures

The following grant information was disclosed by the authors:
Natural Science Foundation of Hubei Province: 2022CFA007.
Hubei University of Technology Ph.D. Research Startup Fund Project: BSQD2020014.

### Competing Interests

The authors declare there are no competing interests.

## Author Contributions

- Jing Tang conceived and designed the experiments, authored or reviewed drafts of the article, and approved the final draft.
- Lun Zhao conceived and designed the experiments, performed the experiments, analyzed the data, performed the computation work, prepared figures and/or tables, authored or reviewed drafts of the article, and approved the final draft.
- Minghu Wu conceived and designed the experiments, authored or reviewed drafts of the article, and approved the final draft.
- Zequan Jiang performed the experiments, authored or reviewed drafts of the article, and approved the final draft.
- Jiaxun Cao performed the computation work, authored or reviewed drafts of the article, and approved the final draft.
- Xiang Bao analyzed the data, authored or reviewed drafts of the article, and approved the final draft.

## Ethics

The following information was supplied relating to ethical approvals (*i.e.*, approving body and any reference numbers):

The Experimental Ethics Committee of Exercise Science of Beijing Sport University granted Ethical approval to carry out the study within its facilities (Ethical Application Ref: 2019007H).

## Data Availability

The code is available at GitHub and Zenodo:

- https://github.com/Lunzo822/DenseNet-LSTM.git

- Lunzo. (2024). Lunzo822/DenseNet-LSTM: SE-DenseNet-LSTM (DenseNet). Zenodo. https://doi.org/10.5281/zenodo.10473549

The raw data are available in the Supplemental Files.

## Supplemental Information

Supplemental information for this article can be found online at http://dx.doi.org/10.7717/peerj-cs.1881#supplemental-information.

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
