# Peer review of "A SE-DenseNet-LSTM model for locomotion mode recognition in lower limb exoskeleton"

_PeerJ Computer Science, doi:10.7717/peerj-cs.1881_

## Round 0.1 · original submission · Major Revisions

Please consider the reviewers' comments to improve the manuscript.

·

Basic reporting

### English language

Generally, the English language is good and on professional level. The word "attention" in the title looks inappropriate.

### Intro, background and references

The abstract and introduction clearly state the problem. In the introduction authors provide a literature survey in which they classify the works according to different methods for human motion capturing, and different methods for recognition. The introduction part of the text is well organized and motivates the presented research.

### Text structure

Mathematical expressions are not typeset very well. It is mainly because MS Word is used to prepare the manuscript. If your text will contain mathematical expressions, it is much better to prepare the article with LaTeX.

Matrix (14) is not written in the standard form. Inline expressions are not typeset correctly. There is problem with the space between operators and operands.

### Figures and tables

Figure 1 is too big. I would suggest to the authors to draw a scheme, instead of using a photograph. This will make the image more informative.

Figure 2 is also too big, and it is blurry. Use vector graphic instead of a raster.

The same holds for Figure 5, Figure 6, Figure 7 and Figure 8.

### Raw data

The authors provided access to their script written in Python.

Experimental design

### Originality of the research and scope of the journal

The research is within the scope of the journal.

### Research questions definition

Research questions are well-defined.

### Technical and ethical standards of the research

Technical and ethical standards of the research are on a high level.

### Description detail sufficiency to replicate

The research is presented in sufficient details.

Validity of the findings

### Impact and novelty

The novelty of the presented approach is clearly stated, along with its limitations.

### Data provided robustness and statistical control

Data used in the research is sufficient to conduct the presented results.

### Conclusions

Well formulated and the method presented in the text is well analyzed.

Additional comments

If the authors correct the typeset of the text and the visual quality of the figures, I believe that the text will be a good paper. If the technical difficulty will not rise too much, I suggest the authors to rewrite the manuscript in LaTeX, since this will fix automatically most of the cosmetic problems in the mathematical expressions.

·

Basic reporting

The paper introduces a hybrid model combining a Dense Convolutional Network (DenseNet) and Long Short-Term Memory (LSTM) with an attention module for recognizing different human locomotion modes. This model is a significant contribution, as it addresses the challenge of accurately recognizing locomotion modes, crucial for seamless interaction between exoskeleton robots and human motion.

Experimental design

The paper thoroughly describes the DenseNet and LSTM components of the model, providing insight into how the model processes and learns from the data. The DenseNet efficiently handles feature extraction, and LSTM addresses long-term dependencies in time series data. The attention module further enhances the model's performance by focusing on the most relevant features​.

Validity of the findings

The model's performance is evaluated using metrics like recall, precision, F1 score, and accuracy. Experimental results show a mean recognition rate of 97% across different locomotion modes.

Additional comments

The paper presents a well-structured, innovative approach with a clear methodology and strong experimental results. The use of DenseNet and LSTM with an attention module for locomotion mode recognition is a noteworthy contribution to the field. The model demonstrates high accuracy and robustness, showing significant potential for practical applications in exoskeleton robots. With some enhancements, particularly in dataset diversity and real-time application testing, this work could substantially impact the development of more intuitive and effective exoskeleton systems.

1. While the current dataset is effective, expanding it to include more subjects with varying physical characteristics could improve the model's generalizability and robustness.

2. The paper could benefit from testing the model in real-time scenarios to evaluate its practical applicability and performance in dynamic environments.

3. Exploring further optimizations in the algorithm to reduce computational requirements could make the model more efficient for real-time applications.

4. A more detailed analysis of the attention module's contribution to the model’s overall performance would provide valuable insights into its functionality.

5. Including comparisons with more recent and advanced models in locomotion mode recognition could provide a clearer picture of the proposed model's standing in the current research landscape.

---

## Round 0.2 · accepted · Accept

The paper can be accepted.

·

Basic reporting

N/A

Experimental design

N/A

Validity of the findings

N/A

Additional comments

The authors have taken into account all my remarks from my previous review. I believe this proposition is a good journal paper in the field of computer science.

·

Basic reporting

N/A

Experimental design

N/A

Validity of the findings

N/A

Additional comments

The authors incorporate the comments accordingly.